# Can Lactose Intolerance Be a Cause of Constipation? A Narrative Review

**DOI:** 10.3390/nu14091785

**Published:** 2022-04-24

**Authors:** Julia Leszkowicz, Katarzyna Plata-Nazar, Agnieszka Szlagatys-Sidorkiewicz

**Affiliations:** Department of Paediatrics, Gastroenterology, Allergology and Paediatric Nutrition, Faculty of Medicine, Medical University of Gdańsk, Nowe Ogrody 1-6, 80-803 Gdańsk, Poland; knazar@gumed.edu.pl (K.P.-N.); agnieszka.szlagatys-sidorkiewicz@gumed.edu.pl (A.S.-S.)

**Keywords:** lactose intolerance, constipation, methane-producing bacteria, breath test

## Abstract

Lactose intolerance and constipation are common in children and impact everyday life, not only for patients but also their families. Both conditions can be comorbid with other diseases or form a part of their clinical presentation, but constipation is not usually associated with lactose intolerance. The typical symptoms of lactose intolerance include abdominal pain, bloating, flatus, diarrhoea, borborygmi, and less frequently nausea and vomiting. In approximately 30% of cases, constipation can be a symptom of lactose intolerance. Constipation is characterized by infrequent bowel movements, hard and/or large stools, painful defecation, and faecal incontinence, and is often accompanied by abdominal pain. This paper provides a narrative review on lactose intolerance, its epidemiology, pathogenesis, the correlation between lactose intolerance and constipation in children, and potential mechanisms of such association.

## 1. Introduction

### 1.1. Lactose and Its Derivatives 

Lactose is a disaccharide, a principal carbohydrate found in the milk of most mammals. Human milk, the leading constituent of the diet of infants for the first few months of life, contains 6.7 wt% of lactose [1] and is the main source of calories in milk [2]. The molecule comprises D-glucose and D-galactose bonded with a β-1,4-glycosidic bond [3].

Lactose, produced from cheese or casein whey, is a common component of infant formula, various dairy products, and pharmaceuticals [1]. Lactose’s synthetic derivative, lactulose, is used in the treatment of functional constipation as a second-line drug [4] due to its osmotic properties. It was also discovered that lactulose plays an important role in infant nutrition as a “Bifidus Factor” that especially enhances the growth of Bifidobacterium in the intestinal tract [5].

### 1.2. Lactose Malabsorption

Lactose is enzymatically hydrolysed in the small intestine into D-glucose and D-galactose before it can be absorbed. A special enzyme, lactase, is needed for the process to occur. Lactase is present on the apical surface of enterocytes in the small intestinal brush border, with the highest expression in the mid-jejunum [6]. Its activity decreases after weaning as a consequence of the normal maturational down-regulation of lactase activity [7] and leads to lactose intolerance—a syndrome resulting in different symptoms upon the consumption of foods containing lactose [8]. The syndrome is one of the most common forms of food intolerance, an example of an adverse food reaction (AFR) caused by a single chemical element within a food [3]. The scheme of enzymatic breakdown of lactose is presented in Figure 1.

Lactose malabsorption can be categorized into four groups, according to the World Allergy Organization (WAO):Developmental lactase deficiency—observed in premature infants due to temporary lactase deficiency which improves with time [6]. In cases of lactase deficiency, lactose is not properly digested (lactose maldigestion) and, therefore, cannot be absorbed in an undigested form (lactose malabsorption) and is fermented by the gut microbiota [10].Congenital lactase deficiency (also called alactasia)—a rare and severe recessive autosomal disorder presenting in infants with severe osmotic diarrhoea at the commencement of feeding, especially present in Finland and Western Russia.Lactase non-persistence (hypolactasia)—the most common condition, defined as the physiological gradual decline of lactase activity after weaning, occurring in about 70% of the global population. Generally, lactase activity in adults constitutes approximately 10% of that of newborns [1].Secondary lactose intolerance—occurs as a consequence of small bowel injury because of conditions such as viral gastroenteritis, giardiasis, celiac disease, or Crohn’s disease [6].

## 2. Epidemiology of Lactose Malabsorption

Nearly two-thirds [6] of the world’s population cannot digest lactose properly. In Asia, Africa, and South America, large parts of the population have difficulties in absorbing lactose, whereas in Europe and North America less than 8% of the population has such problems [11]. 

Table 1 presents the prevalence of lactose intolerance in selected countries, including ones with the lowest and highest proportion of lactose-intolerant individuals as well as countries with the widest variation within the population. Lactose is tolerated best by the populations of Northern Europe, while populations of numerous countries in Africa and Asia are characterised by almost complete lactose intolerance. Some nations are characterized by wide internal variation. Differences between countries or even within regions have a sound basis in history—migration, domestication of cattle, or ethnic composition plays a role in the global distribution of lactose intolerance [12,13]. 

## 3. Symptoms of Lactose Intolerance

Symptoms are closely related to hydrogen excretion and the dose of ingested lactose. Most clinical trials indicate that 12 g of lactose per serving is still well-tolerated by lactose-intolerant individuals [14]. Intolerance to lactose may present with a vast variety of manifestations. The typical symptoms of lactose intolerance include abdominal pain, bloating, flatus, diarrhoea, borborygmi, and less frequently, nausea and vomiting. In some cases, lactose intolerance can present with constipation [7]. Table 2 shows the prevalence of symptoms of lactose intolerance. The mechanism and temporal association of systemic symptoms are still under debate. 

Abdominal pain and bloating are caused by colonic fermentation of unabsorbed lactose [16]. The gut flora provides a salvage pathway for lactose digestion by cleaving lactose into short-chain fatty acids (SCFA) and gas [8]. Unabsorbed lactose in the ileum and colon results in acidification of the colonic contents and an increased osmotic load. That leads to a greater secretion of electrolytes and fluid and a rapid transit time, which results in the appearance of loose stools and diarrhoea [7]. However, this process also has some benefits. The SCFA and other products of fermentation release additional calories from undigested carbohydrates. Moreover, the intestinal microbiota adapts to facilitate the intake of dairy products [2].

Clinical presentation of symptoms depends on a number of factors. Table 3 presents the factors that impact the development of symptoms after lactose digestion.

Research shows that diarrhoea and flatulence during the hydrogen breath test are the most specific symptoms of lactose intolerance, whereas abdominal pain should not be automatically attributed to lactose intolerance even in the presence of lactose malabsorption in children [18]. The variety of symptoms and factors influencing the clinical presentation can be misleading. A randomized, double-blind, crossover trial, evaluating gastrointestinal symptoms in 30 people who reported severe lactose intolerance showed that people who identify themselves as severely lactose-intolerant may mistakenly attribute a variety of abdominal symptoms to lactose intolerance [19].

As the symptoms are subjective, the validated questionnaires regarding symptoms are useful. Sterniste et. al. characterized lactose-induced symptoms with the population-specific, validated paediatric carbohydrate perception questionnaire (pCPQ) and their correlation with the history of symptoms. They stated that symptoms may be more relevant than malabsorption for the clinical outcomes [20]. Moreover, the questionnaires measuring the quality of life show the decline in perceived quality of life of individuals with lactose malabsorption [21].

## 4. Diagnosis of Lactose Malabsorption and Intolerance

There are a few tests available to diagnose lactose intolerance, including hydrogen breath test, quick lactase test, lactose tolerance test, and genetic test. Naturally, along with laboratory diagnostics, attention should be paid to a properly taken history and any abnormalities in physical examination.

### 4.1. Hydrogen Breath Test and Methane Breath Test

The lactose hydrogen breath test is used for the detection of lactose maldigestion, is non-invasive, inexpensive [22,23], and can be used for diagnosis from the first years of life [24]. The method is based on the measurement of the amount of hydrogen produced by bacteria residing in the bowel by taking a breath sample. Some of the hydrogen is expelled in flatus, but most of the gas is absorbed in the large intestine into the bloodstream, and then transported to the lungs and breathed out [22], although some patients’ bacterial flora produces methane instead of hydrogen. Breath methane excretion represents an alternative marker for intestinal gas breath excretion measurement [25]. CH_4_ measurements may possibly be of additional value for the diagnosis of lactose intolerance [23,26,27].

Nevertheless, because of the lack of diagnostic standardization and contradictory results of available tests, reliable and correct diagnosis is still problematic [28]. Schneider et al. suggest that reported symptoms during the breath test are not a reliable method to identify children with carbohydrate malabsorption but might be a measure for carbohydrate intolerance. In general, the approach combining tests and clinical symptoms might help diagnose children with suspected carbohydrate malabsorption [28]. Hammer et al. claim that methane measurement did not significantly affect the detection rate of carbohydrate malabsorbers in children and adolescents with functional abdominal complaints. The use of CO_2_ correction altered the diagnosis of malabsorption in a minority of patients but did not significantly alter overall test results [29]. The North American Consensus stated that the breath test is useful in the diagnosis of carbohydrate maldigestion, methane-associated constipation, and evaluation of bloating or gas but not in the assessment of oro-caecal transit [30].

The breath test has some disadvantages, including the need for a special diet—patients should avoid beans, wheat, and oat flour, potatoes, and corn but favour rice and meat. Children should also fast overnight for 8 to 12 h, and infants <6 months old for 4 to 6 h [31]. Moreover, the test itself lasts long (approximately 3 h) and can trigger gastrointestinal symptoms in patients with lactose malabsorption [32].

### 4.2. Quick Lactase Test

Measuring the activity of lactase, sucrase, maltase, and the lactase-to-sucrase ratio in duodenal biopsies is best for diagnosing hypolactasia [32]. Although, the test does not take into account the development of symptoms. After taking a duodenal sample during a gastroscopy, the biopsy is incubated with lactose on a plate for 20 min and observed for a colour reaction [33]. A quick lactase test is as sensitive as a lactose breath test in detecting lactase deficiency; however, it seems to be more accurate than a lactose breath test in predicting the clinical response to a lactose-free diet [34]. Despite being an invasive procedure (which the biggest disadvantage is the need for anaesthesia in some cases), this test is a good option if gastroduodenoscopy is performed due to other reasons [33,35].

### 4.3. Lactose Tolerance Test

The lactose tolerance test is noncomplex and inexpensive. It involves ingestion of 50 g of lactose and measuring blood glucose at different times (baseline, 30′, 60′, 120′) [36]. A recent study shows that shortening the test to only two measurements (baseline and 30′) leads to hardly any differences in final results [37], making the method even more feasible, although it also has some disadvantages that include inducing potential symptoms (pain, diarrhoea, flatulence, vomiting), its relatively invasive nature (multiple blood draws), and a prolonged duration [37].

### 4.4. Genetic Test

It has been proven that primary lactose intolerance is related to a single-nucleotide polymorphism of the lactase (LCT) gene. In children older than 6 years old, the result of genetic testing based on LCT 13910C > T and 22018G > A polymorphisms may diagnose lactose intolerance [38], especially in Caucasians [2]. Nonetheless, genetic testing does not provide the answer if an individual develops lactose intolerance in the course of life [39].

## 5. Treatment

Current disease management is based on reducing lactose in the everyday diet by avoiding milk and milk-containing products or by drinking milk in which the lactose comes in pre-hydrolysed form [40]. Fermented dairy products, such as yoghurt can be usually consumed because the bacteria present in yoghurt assist with lactose digestion [41]. Evidence has shown that probiotic bacteria in fermented and unfermented milk products can be used to alleviate the clinical symptoms of lactose intolerance [42]. Furthermore, the consumption of yoghurt by those individuals provides the daily recommended calcium and vitamin D intake [43]. They can also drink milk in tolerated amounts (approximately one cup of milk per day) [14]. For patients who are not eager to resign from consuming lactose-containing products, enzyme substitution is a therapeutic option and also leads to improvement of symptoms [31].

Apart from alactasia, it is very important not to exclude lactose from the diet without medical reasons. Suchy et al. described an interesting mechanism regarding misconceptions about being lactose intolerant. It becomes intergenerational: parents with self-diagnosed lactose intolerance place their children on lactose-restricted diets (even if no symptoms are present) in the mistaken belief that the children will develop symptoms if given lactose [40]. It must be taken into consideration that cow’s milk and dairy products are major sources of calcium, phosphorus, choline, riboflavin, vitamin B12, and vitamin A—that is why dairy exclusion could lead to the development of micronutrient deficiencies [44]. In addition, lactose itself enhances the absorption of calcium [45]. Total or partial withdrawal from milk products for a prolonged period is a potential risk factor of defective bone mineralization [46]. Very low intake of vitamin D can lead to the development of rickets [40]. Moreover, the addition of lactose to the diet not only significantly increases the counts of Bifidobacteria and lactic acid bacteria, but also decreases the amount of Bacteroides and Clostridia in healthy infants [47]. The recent RCT study shows that use of selected probiotics (Bifidobacterium longum and Lactobacillus rhamnosus) and vitamin B6 can alleviate symptoms and gut dysbiosis in lactose-intolerant patients [48,49].

## 6. Constipation—Convergences in Differential Diagnosis with Lactose Intolerance

Constipation is characterized by infrequent bowel movements, hard and/or large stools, painful defecation, and faecal incontinence, and is often accompanied by abdominal pain [50]. It is a common problem in the paediatric population, with an estimated prevalence of 3% [5]. Ninety-five percent of cases are accounted for by functional constipation, while the rest (5%) is organic in origin [51]. Among multifactorial pathophysiology of functional constipation, several elements can be distinguished, such as dietary and lifestyle factors (low-fiber diet, obesity, insufficient physical activity or food and fluid intake), positive family history of constipation and parental factors, colonic motility dysfunction, psychological and behavioural disorders [52]. Among organic causes of constipation, significant for this review, celiac disease and food intolerance are mentioned. In the course of celiac disease, secondary lactose intolerance may occur [53], because of damage to the brush border and transient dysfunction of lactase.

### 6.1. Cow’s Milk Allergy

An increasing number of reports suggest a correlation between chronic constipation and cow’s milk allergy in children [54,55,56,57]. There is some evidence that both patients and physicians confuse cow’s milk protein allergy with lactose intolerance [58,59] while the symptoms and treatment are distinctively different, yet still overlapping. IgE-mediated reactions, such as urticaria, angioedema of the oral cavity, or pruritus occur only in the case of cow’s milk protein allergy [3]. Importantly, in lactose intolerance, rectal bleeding is not observed [60]. Nevertheless, the differentiation and final diagnosis should be thorough to establish proper treatment.

### 6.2. Constipation and Gut Microbiota

Irritable bowel syndrome is another diagnosis with symptoms mimicking lactose intolerance, with a subgroup classified as irritable bowel syndrome with constipation (IBS-C). The exact relationship between lactose or fructose maldigestion and IBS is not clear [61]. In their review, Cancarevic et al. stated that despite the fact that IBS patients commonly reported milk intolerance, no conclusive evidence was found to suggest an objective link between IBS and any known malabsorption syndromes, including lactose malabsorption [62]. What is known, constipation-predominant IBS (IBS-C) is associated with increased levels of methanogenic archaea [63]. The gut microbiota has been shown to play a role in IBS-C: compared with healthy subjects, IBS-C patients have a lower level of Actinobacteria, including Bifidobacteria, in their faecal samples and a higher level of Bacteroidetes in their mucosa [64]. It was also shown that selected antibiotics, such as neomycin and rifaximin improve constipation in IBS-C by altering methane production [65,66,67].

There is a suggestion that alternation of the microbiome may lead to constipation by changing the levels of microbial-derived metabolites in the gut [68]. Metabolites of bacterial fermentation can modulate gut functions. Such metabolites include short-chain fatty acids (SCFAs), secondary bile salts (BAs), and methane that could trigger the release of gut hormones from enteroendocrine cells (EECs), such as 5-hydroxytryptamine (5-HT), peptide YY (PYY), and glucagon-like peptide-1 (GLP-1) and modulate gut sensation, secretion, and motility. Research findings were inconsistent. Hence, further studies on this topic need to be conducted [69,70].

### 6.3. Possible Contradictions in Treatment

In view of pharmacological treatment, PEG (macrogol) is the first-choice osmotic laxative in children with functional constipation. If macrogols cannot be administered, lactulose is a drug of choice [50] and is registered for use in younger children [71]. This fact is especially interesting considering the subject of this review. Lactulose, as a derivative of lactose, can possibly trigger symptoms (bloating, flatulence, abdominal cramps, faecal incontinence [72]) of lactose malabsorption, similar to constipation.

There is another shared element in the treatment of lactose intolerance and constipation, namely: probiotics. Oak and Jha’s systematic review showed varying degrees of efficacy but an overall positive relationship between probiotics contained in fermented and unfermented milk products and lactose intolerance [42]. A systematic review by Leis et al. confirmed the overall positive relationship between probiotics and lactose intolerance, but they also noted that further clinical trials were needed in order to gather more evidence [73]. With respect to constipation, probiotics have been suggested as a potential treatment for this condition, yet recent research suggests they are ineffective for the disease management and successful treatment of functional constipation in children [73]. Despite profound research on the role of probiotics on gut motility and constipation, the majority is derived from animal studies, thus further studies on humans are needed to determine the effectiveness of probiotics on constipation [73].

## 7. Methane—A Connection between Lactose Intolerance and Constipation

The major cause of symptoms, especially flatus, in food intolerance, is connected with the production of gases and toxins by gut bacteria. Owing to the low oxygen concentration in the large intestine, >90% of the gut bacteria are anaerobes [74]. Among numerous species residing in the gastrointestinal tract, there are also archaebacteria, responsible for methane production. Several archaebacteria phylotypes of human gastrointestinal tract were identified, including Methaninobrevibacter smithii and Methanospaera stadmagnae, methanogens in the intestine, and the Methannobrevibacter oralis archaeon in the oral cavity [75].

In the low-oxygenated environment, lactose becomes a source of hydrogen (H_2_), methane (CH_4_), and hydrogen sulfide (H_2_S). The H_2_ produced can act as the substrate for the reaction carried out by methanogenic archaebacteria [74]:4H_2_ + CO_2_ → CH_4_ + 2H_2_O

Produced methane possibly acts as a neuromuscular transmitter and delays intestinal transit [75]. It causes the augmentation of contractile activity of the gut and slows the peristalsis down [76]. Studies have shown that patients with chronic constipation have more methanogenic bacteria than healthy subjects [77,78]. Furnari et al. assessed whether different patterns of H_2_ and CH_4_ excretion were related to some intestinal disturbances. They proved that methane excretion was associated with alterations in intestinal motility, particularly common in patients with constipation [79]. A similar outcome was observed by Pimentel et al. Their controlled clinical trial performed on animals and humans with IBS showed that methane slowed intestinal transit and augmented small intestinal contractile activity [80].

The association of breath methane level with the severity of constipation has also been investigated. Recent studies showed that constipation and bloating severity did not correlate with methane levels on a glucose breath test. However, baseline methane levels had stronger correlations with constipation compared with maximum levels of methane achieved [81].

On the other hand, the presence of methanogenic gut bacteria can be beneficial, especially for individuals with malnutrition. It was proved that by removing fermentative hydrogen by methanogens, the more complete oxidation of substrates is possible, which results in improving energy harvest [82,83].

## 8. Rapid-Transit Constipation

As the presence of breath methane [84] or hydrogen [85] in children with chronic constipation may suggest the possibility of prolonged colonic transit time, paradoxically there are also some reports about constipation with rapid colonic transit. This subset of chronic constipation, rapid-transit constipation (RTC), has been linked to food intolerance [86]. It is defined as greater than 25% of tracer beyond hepatic flexure at 6 h and/or greater than 25% of tracer beyond the end of descending colon at 24 h in nuclear transit studies [87]. Children with RTC showed significant improvements in constipation and pain after excluding the sugar indicated by positive breath tests. This outcome suggests that specific sugar-exclusion diets may have a role in the management of RTC in children [86]. This method of treatment corresponds with outcomes of other studies concerning RTC [87,88] but in one case, the exclusion concerned cow’s milk, thus only indirectly concerned lactose.

## 9. Conclusions

Lactose intolerance can present with constipation. Although it predominantly manifests with diarrhoea, approximately one-third present with constipation, especially in hydrogen non-producers. In lactose intolerance, constipation is connected with methane produced by gut bacteria. Diagnosis in this matter is challenging. Hydrogen breath tests are required to detect carbohydrate malabsorption. Additionally, a methane breath test should be performed to support the diagnosis and establish the ratio of H_2_ and CH_4_ in exhaled air. Restricting carbohydrates intake may alleviate the symptoms of constipation, depending on its aetiology. Restricting lactose in the diet can be helpful in some cases, although its presence in the everyday diet is crucial for the proper absorption of nutrients from food and should not be eliminated completely. The composition of gut microbiota plays a crucial role in the clinical manifestation of both lactose intolerance and constipation. Yet, it requires further research. Lactose intolerance coexists with other conditions that have similar symptoms and should be taken into consideration while attempting a differential diagnosis, especially if the patient presents with chronic intractable constipation.

## Figures and Tables

**Figure 1 nutrients-14-01785-f001:**
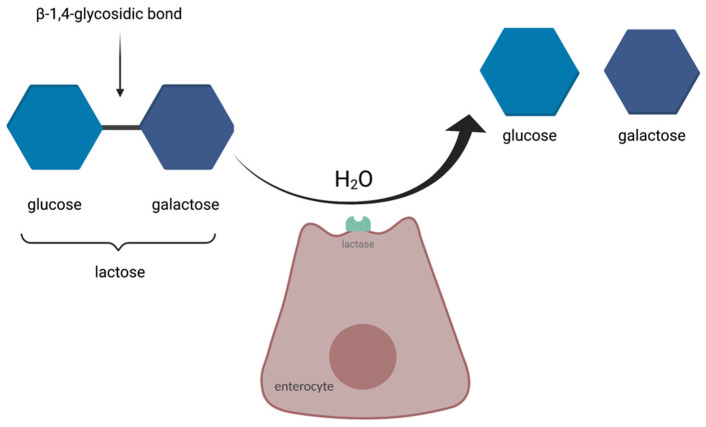
The enzymatic breakdown of lactose, based on [9].

**Table 1 nutrients-14-01785-t001:** Lactose malabsorption or lactase non-persistence prevalence data in adults and children aged 10 years or older, based on [12].

Country	Prevalence (Mean) %	CI of Standard Variation (95% CI)%
Ireland	4	0–8
Denmark	4	0–9
Sweden	7	4–9
United Kingdom	8	7–9
New Zealand	10	8–11
Finland	19	18–20
Poland	43	39–47
Australia	44	35–53
Canada	59	44–74
Russia	61	59–64
Japan	73	59–86
Armenia	98	98–99
Zambia	98	98–99
Vietnam	98	98–99
Solomon Islands	99	98–99
Ghana	100	100–100
Malawi	100	100–100
South Korea	100	100–100
Yemen	100	99–100

**Table 2 nutrients-14-01785-t002:** Symptoms of lactose intolerance, based on [15].

Symptoms of Lactose Intolerance		Frequency (% of Total)
Gut-related	Abdominal pain	~100
	Gut distension	
	Borborygmi	
	Flatulence	
	Nausea	78
	Vomiting	
	Diarrhoea	70
	Constipation	30
Systemic	Headache	86
	Loss of concentration	82
	Long term tiredness	63
	Muscle pain	71
	Joint pain/stiffness	71
	Allergy	40
	Mouth ulcers	30
	Heart arrhythmia	24
	Increased frequency of micturition	<20
	Sore throat	

**Table 3 nutrients-14-01785-t003:** Factors associated with the development of symptoms after lactose digestion, based on [17].

Factors Associated with Development of Symptoms after Lactose Digestion
Lactase deficiency Lactose dosage (≤12 g lactose is usually well-tolerated) Speed of lactose consumption Ingestion with other nutrients Oro-caecal transit time Intestinal microbiota Visceral hypersensitivity Irritable bowel syndrome and other functional gastrointestinal diseases Anxiety, depression, and other psychiatric conditions High levels of psychosocial stress Acute physical stress (e.g., abdominal surgery, infection)

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
