# Peer review of "Can Lactose Intolerance Be a Cause of Constipation? A Narrative Review"

_nutrients, 2022, doi:10.3390/nu14091785_

Round 1

Reviewer 1 Report

Thank you for this nice overview addressing an interesting and undervalued topic. The paper is original, well-structured and generally well-written. However, I do have some comments and suggestions to strengthen the manuscript, as detailed below.

  1. General remark: Please control for correct style of referring (e.g. line 48) and add spaces before the reference if necessary.
  2. Figure 1: please explain the additional value in this paper or remove the figure.
  3. line 34: intestinal tract
  4. line 29-30: please rewrite sentence as it is unclear and difficult to read. 
  5. line 46-47: be carefull not to confuse lactose malabsorption and lactose intolerance (which happens a lot both in general practice and in research). However, down-regulation of lactase activity and malabsorption does not necessarily cause complaints. 
  6. line 59: feeding is more appropiate to say instead of breastfeeding
  7. line 62: "significant" GI symtoms is a strong statement and in my opinion incorrect as research stated that not all people with LM experience symptoms 
  8. line 70-71: The sentence does not seem to fit here. It might be better to add after line 63.
  9. Table 1: please explain the additional value of this table or remove it. 
  10. line 80 (title of table): do you mean lactase-non-persistence? 
  11. table 2: beware that the mechanism and causality/temporal association of systemic symptoms is still under debate. 
  12. table 3: please add a title. It might be more clear to use several columns with different origins (external/GI/...)
  13. Concerning the paragraph about symptoms, I do miss some additional information about the exact mechanism (mention gasses and short chain FA), test/retest reliability, the use of validates symptom registration forms and the concept of colonic salvage. 
  14. line 121: test 
  15. line 128-133: Please rewrite these sentences, as they are rather unclear.
  16. line 139: Hammer 
  17. line 147: Please explain "special diet". Furthermore, the general advised time for fasting in children is 8-12h and 4-6h in infants. 
  18. line 150: although the quick lactase test is a good diagnosis for hypolactasia, it does not take into account the development of symptoms (cfr remark 5)
  19. line 148/166: You mention the duration of the test as a disadvantage of multiple non-invasive tests. However, a gastroscopy (often under narcosis) has also a lot of disadvantages and risks, which seem to be neglected in your text. Furthermore, the provocation of symptoms is in a way a necessary evil to confirm symptoms and thus the diagnosis of LI. 
  20. part treatment: Please also mention the possible use of lactase substitution therapy. 
  21. line 196: replace "of" by "for" 
  22. line 197: I would like you to go more into detail about the role of the microbiome as this plays a key role in both LI and constipation and a lot of research is available
  23. line 230: have instead of has 
  24. line 252-255: Please rewrite this sentence for better comprehension
  25. line 259: I assume you mean > 90% instead of <
  26. line 265-272: extensive biochemical information does not provide additional value. You might better shorten this information or rewrite it in a more physiological way. 
  27. A lot of research is available on the role of methane, both in LI and constipation. Can you please explore this more into detail, as it certainly seems to be the most important association between LI and constipation. 
  28. conclusion: the key messages are ok, however the text is not fluent to read and should be rewritten. 

Author Response

Dear Reviewer,

thank you for valuable comments regarding our manuscript. We agree with all your suggestions. You can find the detailed response below (annotated as “###”).

1.General remark: Please control for correct style of referring (e.g. line 48) and add spaces before the reference if necessary.

### We made necessary corrections.

2.Figure 1: please explain the additional value in this paper or remove the figure.

### After reading all of the remarks, we decided to change the figure. We believe that the new one is more transparent and refers better to the text above.

3. line 34: intestinal tract

We changed this mistake.

4. line 29-30: please rewrite sentence as it is unclear and difficult to read. 

### We rewrote the sentence: “Lactose, produced from cheese or casein whey, is a common component of infant formula, various dairy products, and pharmaceuticals.”

5. line 46-47: be carefull not to confuse lactose malabsorption and lactose intolerance (which happens a lot both in general practice and in research). However, down-regulation of lactase activity and malabsorption does not necessarily cause complaints. 

### We  deleted “in individuals with lactose malabsorption” to make the sentence less misleading.

6. line 59: feeding is more appropiate to say instead of breastfeeding

### We corrected that sentence.

7. line 62: "significant" GI symtoms is a strong statement and in my opinion incorrect as research stated that not all people with LM experience symptoms 

### We deleted the word “ significant”.

8. line 70-71: The sentence does not seem to fit here. It might be better to add after line 63.

### We transferred the sentence after line 63.

9. Table 1: please explain the additional value of this table or remove it. 

### We placed the data of the prevalence of lactose malabsorption in different countries in the table to clearly present the regions of the world where lactose malabsorption is much or less of a problem. We decided that this form of presentation will be the best in this case.

10. line 80 (title of table): do you mean lactase-non-persistence? 

### Yes, we changed this mistake

11. table 2: beware that the mechanism and causality/temporal association of systemic symptoms is still under debate. 

### We added the sentence: “The mechanism and temporal association of systemic symptoms is still under debate”

12. table 3: please add a title. It might be more clear to use several columns with different origins (external/GI/...)

### We added the title. We believe that the table in this form is clear enough.

13. Concerning the paragraph about symptoms, I do miss some additional information about the exact mechanism (mention gasses and short chain FA), test/retest reliability, the use of validates symptom registration forms and the concept of colonic salvage. 

### We restructured the paragraph according to your remarks. We mentioned the mechanism, the use of questionnaires and colonic salvage. The test reliability was mentioned within lines 209-214

14. line 121: test 

### We corrected that mistake.

15. line 128-133: Please rewrite these sentences, as they are rather unclear.

We deleted the sentence: “False-negative results for the H2-breath are observed for ‘H2-non-producers’, the 2–43% of individuals (<10% in most studies) in whom the bowel flora does not produce hydrogen [9].” to make the text more consistent.

16. line 139: Hammer 

### We corrected that mistake.

17. line 147: Please explain "special diet". Furthermore, the general advised time for fasting in children is 8-12h and 4-6h in infants. 

### We added more details to this sentence: “… special diet - patients should avoid beans, wheat and oat flour, potatoes, and corn but favour rice and meat. Children should also fast overnight for 8 to 12 hours, and infants <6 months old 4 to 6 hours”

18. line 150: although the quick lactase test is a good diagnosis for hypolactasia, it does not take into account the development of symptoms (cfr remark 5)

### We added the remark : “Although, the test does not take into account the development of symptoms.”

19. line 148/166: You mention the duration of the test as a disadvantage of multiple non-invasive tests. However, a gastroscopy (often under narcosis) has also a lot of disadvantages and risks, which seem to be neglected in your text. Furthermore, the provocation of symptoms is in a way a necessary evil to confirm symptoms and thus the diagnosis of LI. 

###We ruled out the word “very” as it might be a bit of exaggeration.

We also mentioned the need for narcosis in some cases of gastroscopy. On the other hand, we underlined that this is a good option of diagnosis LI if gastroduodenoscopy is performed due to other reasons.

20. part treatment: Please also mention the possible use of lactase substitution therapy. 

### We mentioned it by adding the sentence: “For patients who are not eager to resign from consuming lactose-containing products, enzyme substitution is a therapeutic option and also leads to improvement of symptoms “.

We also shorten the subtitle to “treatment”, so it would be more accurate than “Treatment – exclusion of lactose from the diet”

21. line 196: replace "of" by "for" 

### We corrected that mistake.

22. line 197: I would like you to go more into detail about the role of the microbiome as this plays a key role in both LI and constipation and a lot of research is available

### We ameliorated the topic but decided to add up the paragraph “6.2. Constipation and gut microbiota” from line 410 to 417

### We added up more details about the role of the microbiome.

23. line 230: have instead of has 

### We corrected that mistake

24. line 252-255: Please rewrite this sentence for better comprehension

### We restructured the sentence.

25. line 259: I assume you mean > 90% instead of <

### We corrected that mistake.

26. line 265-272: extensive biochemical information does not provide additional value. You might better shorten this information or rewrite it in a more physiological way. 

### We shorten the information into this sentence: “ In the low-oxygenated environment, lactose becomes a source of hydrogen (H2),  methane (CH4) and hydrogen sulfide (H2S) [68].”

27. A lot of research is available on the role of methane, both in LI and constipation. Can you please explore this more into detail, as it certainly seems to be the most important association between LI and constipation. 

### We referred to your remarks and ameliorated the indicated topics, mostly in the paragraph called “Methane – a connection between lactose intolerance and constipation”. Only few more information was added, because in our opinion the topic was described comprehensively enough. It was based on the newest research. We believe that we improved the manuscript in the best possible way.

28. conclusion: the key messages are ok, however the text is not fluent to read and should be rewritten. 

### We rewrote the pointed sentences. We would like to mention that English was corrected by two independent certified medical proofreaders.

Thank you for all of the remarks. 

Reviewer 2 Report

In this review, the authors discuss lactose intolerance, constipation, and their relationship with pediatric patients. The review was well-organized and presented. Some minor issues need to be revised.  

Line 32, It  was > It was, additional space.

Line 34, all the bacterial names at family, genus, and species levels should be italic across the manuscript. In addition, the first letter should be capitalized, such as clostridia > Clostridia (line 199).

Line 79, or ethnic composition play > plays.

Line 81, older. Based on [12], a comma should be used before based on. Similar to line 93.

Line 121, breath tes > test.

Line 132 methane (CH4), to list all the abbreviations for the first time shown in the manuscript. In addition, list the full names of NADH and NAD.

Line 142, CO2 > CO2.

Line 259, description of ‘< 90% if the gut microbiota are anaerobes’ is not correct.

Line 287, Section 8. RCT and RTC were used alternatively, correct it.

Author Response

Dear Reviewer, thank you for valuable comments regarding our manuscript. We agree with all your suggestions. You can find the detailed response below (annotated as “###”)

In this review, the authors discuss lactose intolerance, constipation, and their relationship with pediatric patients. The review was well-organized and presented. Some minor issues need to be revised.  

Line 32, It  was > It was, additional space.

### We corrected this mistake.

Line 34, all the bacterial names at family, genus, and species levels should be italic across the manuscript. In addition, the first letter should be capitalized, such as clostridia > Clostridia (line 199).

### We corrected this mistake.

Line 79, or ethnic composition play > plays.

### We corrected this mistake.

Line 81, older. Based on [12], a comma should be used before based on. Similar to line 93.

### We corrected this mistake.

Line 121, breath tes > test.

### We corrected this mistake.

Line 132 methane (CH4), to list all the abbreviations for the first time shown in the manuscript. In addition, list the full names of NADH and NAD.

### In connection with the comments of the second reviewer, we decided to shorten this paragraph and not mention NADH and NAD at all.

Line 142, CO2 > CO2.

### We corrected this mistake.

Line 259, description of ‘< 90% if the gut microbiota are anaerobes’ is not correct.

### We corrected this mistake.

Line 287, Section 8. RCT and RTC were used alternatively, correct it.

### We corrected this mistake.

Thank you for all your remarks.
